# Dealing with Macrophage Plasticity to Address Therapeutic Challenges in Head and Neck Cancers

**DOI:** 10.3390/ijms23126385

**Published:** 2022-06-07

**Authors:** Sonia Furgiuele, Géraldine Descamps, Lorena Cascarano, Ambre Boucq, Christine Dubois, Fabrice Journe, Sven Saussez

**Affiliations:** 1Department of Human Anatomy and Experimental Oncology, Faculty of Medicine, Research Institute for Health Sciences and Technology, University of Mons (UMONS), Avenue du Champ de Mars, 8, 7000 Mons, Belgium; sonia.furgiuele@umons.ac.be (S.F.); geraldine.descamps@umons.ac.be (G.D.); lorena.cascarano@uclouvain.be (L.C.); ambre.boucq@student.umons.ac.be (A.B.); fabrice.journe@umons.ac.be (F.J.); 2Cytometry Core Facility, Université Libre de Bruxelles (ULB), 1050 Brussels, Belgium; christine.dubois@ulb.be; 3Laboratory of Clinical and Experimental Oncology, Institut Jules Bordet, Université Libre de Bruxelles (ULB), 1070 Brussels, Belgium; 4Department of Otolaryngology and Head and Neck Surgery, CHU Saint-Pierre, 1000 Brussels, Belgium

**Keywords:** HNSCC, macrophage polarization, THP1, PBMC, M1 and M2 macrophages, tumor microenvironment, cancer treatment, metabolism, glutaminolysis, oxidative stress

## Abstract

The head and neck tumor microenvironment (TME) is highly infiltrated with macrophages. More specifically, tumor-associated macrophages (TAM/M2-like) are one of the most critical components associated with poor overall survival in head and neck cancers (HNC). Two extreme states of macrophage phenotypes are described as conducting pro-inflammatory/anti-tumoral (M1) or anti-inflammatory/pro-tumoral (M2) activities. Moreover, specific metabolic pathways as well as oxidative stress responses are tightly associated with their phenotypes and functions. Hence, due to their plasticity, targeting M2 macrophages to repolarize in the M1 phenotype would be a promising cancer treatment. In this context, we evaluated macrophage infiltration in 60 HNC patients and demonstrated the high infiltration of CD68+ cells that were mainly related to CD163+ M2 macrophages. We then optimized a polarization protocol from THP1 monocytes, validated by specific gene and protein expression levels. In addition, specific actors of glutamine pathway and oxidative stress were quantified to indicate the use of glutaminolysis by M2 and the production of reactive oxygen species by M1. Finally, we evaluated and confirmed the plasticity of our model using M1 activators to repolarize M2 in M1. Overall, our study provides a complete reversible polarization protocol allowing us to further evaluate various reprogramming effectors targeting glutaminolysis and/or oxidative stress in macrophages.

## 1. Introduction

In 2018, approximately 18 million cases of cancer were diagnosed worldwide causing 9.6 million deaths. Head and neck cancers are still the sixth most frequently occurring cancers in the world [1]. Head and neck squamous cell carcinomas (HNSCC) are heterogenous tumors encompassing the oral cavity, pharyngeal (oro-, naso-, hypo-), and laryngeal cancers. Often diagnosed at advanced stages, these cancers are frequently associated with an unfavorable prognosis despite the constant and significant evolution of therapeutic strategies; the 5-year survival rate is around 50% and recurrences occur in 40–60% of treated patients [2]. Clinicians are therefore constantly looking for new prognostic biomarkers to better predict the aggressiveness of these cancers, but also for predictive biomarkers in order to select the patients that are most likely to respond to new therapies. HNSCC are mostly associated with alcohol and tobacco consumption but also with human papillomavirus (HPV) infection, and they are less reported to diet [3,4,5].

Treatment response and tumor progression are governed by the interaction between cancer cells and the TME [6]. This TME is mainly composed of immune cells such as macrophages, lymphocytes, dendritic cells and natural killer cells, which will guide tumor development [7]. In fact, among the events leading to the poor prognosis of HNSCC, the TME composition is a critical one because it is well reported that high CD68+ macrophages, low CD8+ T-lymphocytes (LT) and low FoxP3+ T-regulatory lymphocyte (LTreg) infiltrations are associated with the lowest rate of survival [8,9,10,11]. More precisely, these cancers are highly infiltrated by tumor-associated macrophages (TAMs) whose expression characterized by the CD163 marker is correlated with poor overall survival (OS) [12,13,14,15]. In HNSCC, HPV infection induces a specific TME composition that we previously reviewed [16]. Moreover, radio-resistance represents a major problem in the treatment of HNSCC and it seems that TME factors may contribute to this phenomenon. Indeed, it has been recently demonstrated that M2-like macrophages, meaning TAMs, secreted growth factors leading to the radio-resistance in HPV-negative HNSCC, indicating the major role played by macrophages in cancer progression [17].

Despite greater complexity, macrophages continue to be commonly classified as M1 or M2 macrophages. These extreme phenotypes depend on specific marker expression and activities limiting tumor progression and promoting T-helper 1 (Th1) type responses or promoting tumor and supporting T-helper 2 (Th2) type responses, respectively, for M1 and M2. These two classifications do not reflect the reality of in vivo conditions. Indeed, they are the extreme opposing ends of a whole spectrum of possible macrophage polarization states (M1, M2a, M2b, M2c) [18]. Concerning, the M1 phenotype, it triggers an antitumor response via the secretion of pro-inflammatory cytokines (IL-1β, IL-6, IL-12, IL-18, IL-23) and its phenotypic profile is characterized by the expression of MHCII, CD68, CD80, and CD86 [19]. By contrast, the M2 phenotype (associated with TAM) is involved in tumor progression through the production of anti-inflammatory cytokines (IL-4, IL-10, IL-13, TGFβ) and chemokines that allow the recruitment of cells that induce a Th2 response. Phenotypically, M2 macrophages express the macrophage mannose receptor (MMR), corresponding to CD206, as well as CD200R, CD163, MGL1 and MGL2 [20]. In addition, the secretion of immunosuppressive cytokines prevents the defense exerted by cytotoxic T cells which are necessary for the elimination of cancer cells [21]. Furthermore, M1 and M2 macrophages have distinct metabolic states and differentially respond to reactive oxygen species (ROS) production [22]. M2 is also able to inhibit its M1 counterparts. Indeed, M1 and M2 phenotypes are dynamic and show strong plasticity. In this way, an imbalance of the polarization process between M1 and M2 may be a step toward initiating certain pathologies, including cancer [23]. Indeed, tumor progression is associated with an increase in the M2–M1 ratio and is therefore related to a poor therapeutic outcome [24,25]. In this context, an innovative cancer treatment strategy would be the reprogramming of M2 in M1 macrophages in order to create a more efficient anti-tumoral microenvironment which would also be more radiosensitive.

In the present study, we first quantified the macrophages infiltration in 60 patients suffering from HNSCC. Then, we optimized an in vitro protocol for the polarization of monocytes into M0, M1, and M2 macrophages by studying some specific gene and protein expressions. Indeed, polarization methods are highly controversial in the literature [26,27,28]. Moreover, we particularly investigate the glutaminolysis pathway which is favored in M2 macrophages. Alongside their metabolic differences, macrophages also raise different levels of oxidative stress. M1 macrophages, due to their pro-inflammatory functions, express many more ROS compared to the anti-inflammatory ones. Finally, we aim to examine the plasticity of our polarized macrophages from a perspective to study new drugs that reprogram M2 macrophages to an M1 phenotype. These findings open the way to new therapeutic strategies by targeting the pathways involved in both cancer cells death and in the immune system [29].

## 2. Results

### 2.1. Macrophage Characterization in the HNSCC TME

#### 2.1.1. Clinical Cohort Composition

Our clinical series included a total of 60 HNSCC patients, among which 43 (71.7%) were men and 17 (28.3%) were women, with a median age of 62 years old (range, 42–89). Among these patients, 26 patients presented tumor recurrence and 28 died. HPV infection was demonstrated through p16 expression in 29 patients. Table 1 describes the clinicopathological characteristics.

#### 2.1.2. Macrophages Infiltration in a Clinical Series of 60 HNSCCs

In the HNSCC surgical specimens, macrophages were detected by using specific antibodies against CD68, CD80 and CD163. The total macrophages (CD68+), M1 phenotype macrophages (CD80+) and M2 phenotype macrophages (CD163+) were counted in five random fields (magnification 400×, 0.181 mm^2^ area) in both stromal (ST) and intratumoral (IT) compartments (Figure 1a–c).

In order to quantify the percentage of macrophages infiltration in the tumor patients, ratios were calculated between the number of CD68+ cells and the total number of cells by field (0.1 mm^2^). On average, there were 20% (range, 0–62%) of CD68+ in ST and 9% (range, 0–35%) of CD68+ in IT. Of note, the correlation between CD68+ and the addition of CD80+/CD163+ cells was evaluated (Spearman correlation = 0.43, *p* = 0.001). However, a significant difference was observed between the two groups. In fact, there were 6.11-fold more CD80+/CD163+ than CD68+ macrophages. The hypothesis is that some macrophages expressed double labeling. Thus, the number of each phenotype, M1 or M2, was evaluated by dividing the number of specific macrophages by the number of total macrophages (M1 + M2). Then, the percentage of each infiltration throughout the tumor was determined. Our results highlight a significant increase in M2 macrophages compared to M1 macrophages in the total tumor (i.e., intra-tumoral and peri-tumoral, 63%, *p* < 0.0001) (Figure 1d) but also in the ST compartment (62%, *p* < 0.0001) (Figure 1e) and in the IT compartment (61%, *p* < 0.0001) (Figure 1f).

### 2.2. THP1 Monocyte Differentiation in M1 and M2 Macrophage Phenotypes

#### 2.2.1. Analysis of Morphological Changes during Macrophage Polarization

The THP1 monocytes were subjected to polarization into M0, M1 and M2 macrophages thanks to the protocol described in the Materials and Methods section. This protocol was optimized in order to try to reprogram M2 in M1 macrophages. Pictures of the THP1 cell line as well as of the different subtypes of polarized macrophages were taken to observe the morphological changes of cells during the polarization protocol. Since THP1 are monocytes, they are small cells with a rounded and regular morphology. We also saw that the majority of M0 retained a rounded appearance, while others were characterized by an enlarged amoeboid cell shape. Indeed, the M1 types are similar to M0 with the presence of short cytoplasmic extensions. Concerning M2 macrophages, they are more heterogeneous, but the most remarkable feature is the presence of long cytoplasmic extensions (white arrows) (Figure 2a). Next, flow cytometry highlights that the polarization of monocytes into M0 macrophages leads to an increase in cell size (FSC, forward scatter) and granularity (SSC, side scatter). These characteristics can be plotted in size (FSC, x axis) versus granularity (SSC, y axis) of cells (Figure 2b).

#### 2.2.2. Gene Expression Variations during the Macrophage Polarization

To further characterize the THP1 monocytes and polarized macrophages, we quantified the gene expression of different macrophage markers in THP1, M0, M1 and M2 macrophages. *SOCS1* (suppressor of cytokine signaling 1) is involved in the negative regulation of cytokines that act via the JAK/STAT pathway and it is also known as a tumor suppressor and M1 marker. As shown in Figure 3a, *SOCS1* expression is significantly increased in M1 macrophages compared to THP1 monocytes (*p* = 0.008), M0 macrophages (*p* = 0.007) and M2 macrophages (*p* = 0.016). Another interesting M1 marker is *IL-12* (interleukin 12), a pro-inflammatory cytokine which is upregulated after LPS/IFN-γ stimulation with a significant difference between THP1 and M1 (*p* = 0.021), M0 and M1 (*p* = 0.024), and M2 and M1 (*p* = 0.05) (Figure 3b). *PD-L1* (programmed death-ligand 1) also appeared to be significantly increased in M1 macrophages compared to THP1, M0 and M2 (*p* = 0.04, *p* = 0.04, *p* = 0.041, respectively) (Figure 3c). Concerning THP1 polarization in M2 macrophages, we evaluated the expression of *CD206* (cluster of differentiation 206), also known as the mannose receptor, and *CCL2* (chemokine ligand 2), also known as MCP1 (monocyte chemoattractant protein 1), both being M2 markers. *CD206* mRNA is significantly upregulated in M2 compared to THP1 (*p* = 0.008), M0 (*p* = 0.01) and M1 (*p* = 0.015) (Figure 3d). Regarding *CCL2*, its mRNA expression is significantly increased in M2 versus THP1 (*p* = 0.021) and M0 (*p* = 0.023) but no difference appears in comparison to M1 (*p* > 0.05) (Figure 3e). Finally, *CD68* is a widely used marker for highlighting macrophages. *CD68* mRNA is well expressed in all subtypes of macrophages with a significant difference between THP1 monocytes and M1 macrophages (*p* = 0.044) (Figure 3f).

#### 2.2.3. Differential Expression of M1 and M2 Specific Proteins

Moreover, we evaluated our polarization model of THP1 cells by examining the proteins that are known to be specifically expressed by the different macrophage subtypes. First, we performed a series of immunofluorescences targeting intracellular and membrane markers. CD68 is located in the intracellular part and is essentially present at the level of the membrane of lysosomes/endosomes. This protein is weakly expressed in THP1 cells and its expression highly increases during their polarization into macrophages, independently of the final phenotype (M1 or M2) (Figure 4a). The CD14 protein plays an important role in the process of the phagocytosis of macrophages as well as in the recognition of bacterial antigens. This marker is highly expressed in monocytes and to a lesser extent in polarized macrophages (Figure 4b). CD36 is a receptor located on the surface of phagocytic cells. Its expression increases on the surface of M1 and M2 macrophages compared to M0 and monocytes (Figure 4c). The CD80 and CD86 proinflammatory markers are solely expressed in M1 macrophages (Figure 4d,e). By contrast, CD163 and CD206 are two markers that are widely used in order to highlight type M2 macrophages (Figure 4f,g).

Secondly, the protein expression of CD68 was also determined by flow cytometry for quantitative evaluation. The results obtained agree with those observed by immunofluorescence or by RTqPCR. Indeed, the expression of CD68 is detected in almost all subtypes of cells (monocytes or macrophages). However, the fluorescence intensity that was detected by FACS showed that M1 and especially M2 detached from monocytes and M0 with a stronger expression in M1 and M2 subtypes (Figure 5a). In addition, dot plots in Figure 5b–d combine the expression of two different markers (one on the abscissa and the other on the ordinate) and show different cell populations distributed in quadrants: single positives which express only one of the two markers (Q1 and Q4), double negatives which express nothing (Q3), and double positives which express both markers simultaneously (Q2). Around these graphs, histograms referring to each marker (top histogram for the *X*-axis marker and the right one for the *Y*-axis marker) provide the same information but with a different visual approach (in the form of a peak). Concerning M1 macrophages with CD68/CD86 combination markers, double positive cells are observed at 70.12%. Moreover, 29.3% of cells are CD68+/CD86− while only 0.11% of cells are CD68−/CD86+ indicating that almost all CD86+ cells are CD68+ macrophages (Figure 5b). However, for M2 markers, CD68+/CD163+ is only expressed by 18.4% (Figure 5c) and CD68+/CD206+ by 17.5% of cells (Figure 5d). Cutoffs were defined regarding the negative control profile (sample without antibody) for each condition.

### 2.3. Validation of Macrophage Polarization Process on PBMC Differentiation

After monocytes isolation from PBMCs by using CD14+ microbeads, the polarization protocol was applied to PBMC-derived monocytes and validated by immunofluorescence with the three main markers: CD68 (macrophage lineage), CD86 (M1) and CD206 (M2). M1 and M2 macrophages polarized from PBMC are both CD68+ cells (Figure 6a). Moreover, CD86 marker is more intensely expressed by M1 than M2 macrophages as some M2 cells exhibit weaker immunostaining (Figure 6b). Regarding CD206 expression, we observe in Figure 6c that this marker is abundant in M2 macrophages and occasionally detected with a slight expression in some M1 macrophages. Altogether, these data are similar to those shown in Figure 4e,g, and indicate that the polarization protocol we developed using THP1 cells also works using PBMC-derived monocytes.

### 2.4. Characterization of M1 versus M2 Macrophage Phenotype

#### 2.4.1. Metabolism Variations

As it is known that subtypes of macrophages may present distinct metabolic profiles (glycolysis in M1 and glutaminolysis in M2), we investigated the specific actors of glutamine metabolism by assessing *KGA* (kidney-type glutaminase, *GLS1*) and *SLC1A5* (glutamine transporter) by RTqPCR. Our data showed that these two factors are more expressed in M2 macrophages compared to M1 (*p* = 0.005 and *p* = 0.059 for *KGA* and *SLC1A5*, respectively) (Figure 7). Therefore, M2 macrophages demonstrate a higher expression of glutaminase 1 and transporter of glutamine, although the latter is borderline significant, and consequently a stronger glutamine metabolism.

#### 2.4.2. Oxidative Stress Comparison

We next evaluated the oxidative stress differences in our polarization model. First, the gene expression of *NOX2* (NADPH oxidase), which induces O_2_^•−^ formation, and *SOD2* (superoxide dismutase), which induces H_2_O_2_ formation, were quantified by RTqPCR. As shown in Figure 8a, *NOX2*, as well as *SOD2* mRNA expression are higher in M1 macrophages compared to M2 macrophages (*p* < 0.001 and *p* = 0.003, respectively). Moreover, we validated that the M1 population expressed higher levels of ROS than the M2 population (*p* = 0.001), and these data were obtained with the Muse^®^ Oxidative Stress Kit (Austin, TX, USA) (Figure 8b). Then, we investigated Nrf2 (Nuclear factor 2) expression in the two types of macrophages. Anti-Nrf2 immunofluorescence showed a clear increased expression in nuclei of M1 macrophages, in opposition to M2, highlighting the initiation of anti-oxidative mechanisms and the management of stress due to the presence of high levels of ROS in these M1 cells (Figure 8c).

### 2.5. Macrophage Editing as a Target for Cancer Therapy

Finally, in order to point out macrophage plasticity, M2 macrophages were treated for 24 h with M1 activators (LPS + IFN-γ). The different markers evaluated to identify a possible switch were those that emerged as significant or expressing a trend in our previous analyses. After M2 treatment with LPS and IFN-γ, *SOCS1*, *NOX2* and *SOD2* were significantly upregulated in M2-treated compared to M2-untreated (*p* < 0.001, *p* = 0.002 and *p* < 0.001, respectively) and, for *SOCS1* and *NOX2*, reached the M1 values (Figure 9a–c). Indeed, we observed that there was no difference between *SOCS1* and *NOX2* expression in M2-treated compared to their expression in M1. In addition, *CD206* and *KGA* upregulated in M2 were significantly decreased when M2 was treated with M1 activators (Figure 9d,e). Of note, the treatment of M2 with LPS and IFN-γ did not allow decrease *CD206* and *KGA* values to those found in M1 as the difference between M1 and M2-treated was always significant. Eventually, the mRNA expression of the glutamine transporter, *SLC1A5*, significantly increased in M2 after 24 h of treatment with M1 activators compared to M2-untreated (*p* = 0.03). The difference stays significant between M2 treated and M1 macrophages (*p* = 0.004) (Figure 9f). Hence, M2 exposure to M1 activators leads to the repolarization of M2 to M1 macrophages and demonstrates the possible dynamic changes which are allowed by our model.

## 3. Discussion

Macrophage study in a cancer environment has aroused great scientific interest over the last past 10 years. Indeed, besides cancer cells, macrophages, as all components of the TME, play a critical role in tumor progression. HNSCCs present a distinct TME with a low infiltration of CD8+ LT and Foxp3+ LTreg and a high infiltration of TAMs CD68+/CD163+ (related to M2-like macrophages) which are associated with patients’ shorter overall survival [8,9,11,30]. Being the most abundant immune cell type and a key player in the development of HNSCCs, this work focused on macrophage phenotype in HNSCCs and their polarization using in vitro models. In fact, their differentiation depends on cytokine exposure expressed in the TME and may be basically classified as pro-inflammatory/anti-tumoral M1 and as anti-inflammatory/pro-tumoral M2 macrophages. In our clinical study enrolling 60 HNSCC patients, we showed a high infiltration of CD68+ cells mainly composed of CD163+ M2 macrophages (more than 60%) in tumor patients. Of note, we observed that the number of CD68+ cells was lesser than the sum of CD80+ and CD163+ macrophages. This observation suggests that some macrophages may express both M1 and M2 markers because they are in an intermediate phase of the phenotype switch [31,32]. Indeed, an in vitro study has shown that stimulating the polarization of macrophages with LPS + IFN-γ and IL-4 + IL-13 induces the co-expression of the CD86 and CD206 markers by the cells [33]. The same phenomenon probably occurs within the tumor since TAM do not just belong to M1-like or M2-like phenotypes during the whole process of tumor progression. At the early tumor stage, TAM are M1-like phenotypes before switching to the M2-like type [34]. Our study highlights that HNSCCs are more significantly infiltrated with M2 than M1 macrophages. This phenotype status may be explained by the influence of oral microbiota on the TME composition by modulating the immune system [35]. This phenomenon is well reported in colorectal cancer where macrophage polarization is influenced by the gut microbiota composition [36,37]. In HNSCCs, it has been demonstrated that *Fusobacterium periodonticum*, *Parvimonas micra*, *Streptococcus constellatus*, *Haemophilus influenza* and *Filifactor alocis* are up-regulated and that *Streptococcus mitis*, *Haemophilus parainfluenzae*, *Porphyromonas pasteri*, *Veillonella parvula* and *Actinomyces odontolyticus* are down-regulated gradually from stage 1 to stage 4 in oral squamous cell carcinoma patients compared to healthy donors, suggesting a potential role of the oral microbiota in cancer progression [38]. In addition, drinking alcohol is a well-known risk factor in the development of HNSCC and may also influence the bacterial composition of the oral microbiome [39]. An interesting immunomodulatory strategy would be to modulate macrophage phenotypes through the administration of probiotics. The key of such an innovative therapeutic strategy must be to switch M2 to M1 macrophages by targeting the oral microbiota.

Therefore, we then set up a polarization protocol. We initially advocated the use of THP1 monocytes and then validated our data using patient-derived PBMC. Indeed, due to the limited amount of PBMC and their high genetic variability, THP1 cells have been used to overcome such experimental limitations [40]. The first step relies on the generation of M0 macrophages. Currently, the literature remains very controversial regarding the best method to achieve the in vitro polarization of macrophages. PMA is widely used for the induction of monocyte-to-macrophage differentiation [26]. However, the PMA concentration can vary by 80-fold (from 6–500 nM) among studies while the stimulation period can range from 3 to 72 h [27]. In addition, recent protocol updates revealed that 72 h is the best rest timing (exposure without PMA), as a shorter period creates difficulty in moving M0 macrophages towards the M2 phenotype [28].

It should be kept in mind that macrophages are cells with high plasticity. Therefore, changes in their environment lead to rapid modifications of their phenotype. In this way, they adopt biological characteristics approaching either M1 or M2 [41]. In order to generate M1 macrophages, we used 20 ng/mL IFN-γ and 10 pg/mL LPS. IFN-γ is a key cytokine produced by activated T cells and it activates an innate response by upregulating inflammatory cytokines which modulate the TME [42,43]. Concerning the second activator, it is reported that more than 10 pg/mL LPS increases cytotoxicity [44]. After LPS + IFN-γ stimulation of M0 macrophages, *SOCS1*, *IL-12* and *PD-L1* gene levels were upregulated as markers of M1. *SOCS1* expression is induced by LPS and acts on the JAK/STAT pathway. More specifically, SOCS1 inhibits JAK1 and JAK2 by blocking substrate binding, using the JAK kinase inhibitory region [45,46]. In addition, *IL-12* mRNA expression also reflects the effective M1 polarization. IL-12 is a pro-inflammatory chemokine inducing a Th1 response [47,48]. We have also highlighted an increase in expression of the immune checkpoint *PD-L1* after LPS + IFN-γ exposition. Interestingly, the study of Oguejiofor et al. showed a better prognosis of HPV-negative oropharyngeal squamous cell carcinoma patients when their tumors presented a higher infiltration of CD68+/PD-L1+ macrophages within the stroma compartment. However, they did not correlate CD68+ macrophages to M1 or M2 macrophages [49]. In parallel to these specific gene variations, we also investigated M1-related protein expression. Indeed, we demonstrated the overexpression of CD80 and CD86 proteins in M1 compared to the M2 phenotype. These two receptors are well documented as overexpressed in M1 macrophages [50,51]. To prove the efficacy and the yield of our protocol, we observed that 73.2% of M1 were CD68+/CD86+ using FACS analysis.

Simultaneously, M2 macrophages were differentiated from M0 with 20 ng/mL IL-4 and IL-13 for 72 h [44]. These interleukins are involved in the Th2 response and act on the JAK/STAT signaling pathway [52]. CCL2 is a chemo-attractive molecule secreted by both HNSCC cells and macrophages to recruit TAM and inhibit CD8+ LT, respectively [13]. As for the M1 phenotype, several specific protein expressions were evaluated in M2. CD206 and CD163, known, respectively, as scavenger and mannose receptors are two widely used markers to identify M2 macrophages [44,51,53]. CD163 induces Th2 cytokines, however, its exact role in TME remains unclear [54]. Similarly, the immune role of CD206 remains poorly documented in the literature. Our analyses by RT-qPCR and/or by immunofluorescence of these markers clearly showed a significant membrane expression of CD206 and CD163 proteins in M2 macrophages compared to M1 macrophages. However, the percentages of CD68+/CD206+ and CD68+/CD163+ cells that were detected by the FACS analysis were less than 20% of the population. One explanation for this weak yield of polarization would be that not all of the M0 cells reacted to the IL-4 and IL-13 stimuli and therefore did not polarize into M2. The same results were observed in another study [44]. This suggests that additional polarization factors are needed for complete M2 differentiation. Additionally, our polarization protocol was also validated on monocytes that were isolated from PBMCs. These results strengthened the validity of our polarization method and its possible use with patient-derived monocytes.

Along with M1 and M2 specific markers, we also compared CD68, CD14 and CD36 expression during the polarization process. CD68 is a widely used marker for highlighting monocytes and macrophages. Indeed, this intracellular marker that is mainly present at the membrane of lysosomes/endosomes is highly expressed in the monocyte/macrophage lineage, independently of the phenotype. In our study, we observed the presence of this marker in THP1 and in each subtype of macrophages (M0, M1 and M2) at a higher level in M1 and M2. This observation was confirmed by RT-qPCR, immunofluorescence and FACS. CD14 is a membrane-associated protein that forms a highly sensitive LPS signaling complex with TLR4 (Toll-Like Receptor 4) and MD2, and is essential for inflammatory gene expression [55]. CD14 decreases with macrophage differentiation as we validated by immunofluorescence and as it was observed in previous studies after PMA exposure [44,56]. Concerning the CD36 receptor, we found that the expression of this marker was increased on the surface of macrophages compared to monocytes. In fact, Genin et al. also used this marker to validate their polarization method after PMA treatment. They showed, as in our study, a greater expression of CD36 on the surface of M1 and M2 compared to THP1 by immunofluorescence [44].

Furthermore, there is a growing interest in the understanding of the metabolism between cancer cells and the surrounding stromal cells which have been described as exhibiting a distinct metabolic profile [22]. This metabolic plasticity is also a major feature found in macrophages. Classically, M1 capture their energy from glycolysis, fatty acid synthesis, and the pentose phosphate pathway, whereas M2 make extensive use of glutaminolysis and fatty acid oxidation to fuel oxidative phosphorylation [22,57]. A recent study reported that glutamine deprivation prevented IL-4 mediated polarization, highlighting the importance of glutaminolysis metabolism for M2 polarization [58]. Therefore, besides their opposite phenotype, M1 and M2 macrophages exploit distinct metabolism profiles, and this was also confirmed in our study. In fact, *KGA* and *SLC1A5* mRNA were more expressed in M2 macrophages compared to M1. These results correlated with the literature [59]. Indeed, SLC1A5, a glutamine transporter, and KGA (human kidney-type glutaminase), a glutaminase isoform involved in the hydrolysis of glutamine to glutamate, are highly expressed in M2 macrophages because they participate in the glutaminolysis that is mainly used by M2. Optimizing these metabolic changes is crucial for macrophage polarization. Indeed, disruption of these pathways prevents functional polarization and may therefore lead to the repolarization of macrophages [60]. In this context, some studies have shown that M2 macrophages can be repolarized into pro-inflammatory M1 by influencing metabolic mediators [61]. Indeed, inducing the reprogramming of M2 to M1 phenotype could be established following modification of macrophage metabolism by drugs that are clinically approved or under medical follow-up. For example, the activity of a drug such as CB-839 (Telaglenastat) targeting the glutaminase could be of interest for reprogramming macrophages towards a pro-inflammatory phenotype. Indeed, thanks to the previous pre-clinical studies, Telaglenastat is already in phase I/II clinical trials with a favorable safety profile [62,63] because this molecule may also act simultaneously on cancer cells.

Another major difference is observed at the level of the electron transporters NADPH and FADPH which feed the mitochondrial electron transport chain in M2, leading to the production of ATP. Contrariwise, in M1, the electron transport chain tends to increase the production of ROS [64]. The oxidative stress pathways are important for macrophage polarization since the production of ROS is necessary for the activation and functioning of M1, while they are also involved in the differentiation towards TAM, consequently promoting tumor development [65]. In our study, we have demonstrated an elevation of ROS in M1 macrophages compared to M2. Modulating ROS levels within the tumor to reprogram macrophages while targeting cancer cells also seems to be a relevant alternative treatment. This will be performed by using iron oxide nanoparticles. As described by Rojas et al., the treatment of macrophages with superparamagnetic iron oxide nanoparticles (SPIONs) induces an increase in ROS as well as ferroportin expression, reprogramming macrophages towards a pro-inflammatory phenotype [66]. Other groups demonstrated that macrophage treatment with coated SPION increased CD80 and CD86 expression on macrophages [67]. Likewise, Ferumoxytol, an FDA approved treatment of iron deficiency, increases the M1 macrophage phenotype which induces cancer cells apoptosis [68,69].

Thus, macrophages have emerged as a promising therapeutic target in the field of new cancer treatments due to their strategic involvement in the TME. Because of their great plasticity, macrophages can be reprogrammed according to specific signals in order to pass from a pro-tumor phenotype to an anti-tumor one, and then to oppose cancer progression [29,60,70]. Therefore, we aimed to confirm the macrophages plasticity with our polarization model and found an upregulation of M1 markers in M2 treated macrophages. With these last results, we highlight the achievement of a complete polarization protocol from monocytes to macrophages but also demonstrate the possible reversibility of our model, allowing us to further evaluate various repolarization effectors.

## 4. Materials and Methods

### 4.1. Patients and Clinical Data

Sixty patients presenting HNSCC were enrolled in our study (Table 1). Formalin-fixed paraffin-embedded (FFPE) specimens obtained after surgical resection at Saint-Pierre Hospital (Brussels, Belgium) and Jules Bordet Institute (Brussels, Belgium) between 2010 and 2019 were used for immunohistochemical labeling. This retrospective study was reviewed and approved by the Ethics Committee of Jules Bordet Institute (number CE2319, 15 January 2015).

### 4.2. Immunohistochemistry

The 5 μm thick slices of HNSCC were deparaffinized in toluene and rehydrated in a graded series of alcohol baths, then peroxidase activity was inhibited using 3% H_2_O_2_ in distilled water and finally, the slices were rinsed with water for 7 min. Antigen retrieval was processed by immersing the samples in 10% EDTA/H_2_O (for anti-CD68) or in 10% citrate/H_2_O (for anti-CD80 and anti-CD163) followed by heating for 6 min in a pressure cooker. Non-specific sites were blocked with 0.5% casein for 15 min (for anti-CD68) or 1 h (for anti-CD80 and anti-CD163). Then, the slices were incubated with primary antibody: anti-human CD68 monoclonal mouse, dilution 1:200, from Dako (Uden, The Netherlands); anti-human CD80 monoclonal mouse, dilution 1:20, from R&D (Minneapolis, MN, USA); and anti-human CD163 monoclonal mouse, dilution 1:50, from Sanbio (Santa Clara, CA, USA) for 1 h at room temperature (anti-CD68 and anti-CD163) or overnight at 4 °C (anti-CD80). Finally, the BrightVision Poly-HRP IgG kit was used for the second antibody and the antigens were consequently visualized through the addition of diaminobenzidine and H_2_O_2_. For each experiment, tonsil tissue was used as positive (and negative—no primary antibody) controls. The number of each immune cell type was counted in 5 fields in the IT and ST compartments with an Axio-Cam MRC5 optical microscope (Zeiss, Hallbergmoos, Germany) at 400× magnification by two investigators (S.F. and G.D.). The mean of each counting in the 0.181 mm^2^ area was calculated for each patient.

### 4.3. Cell Culture

THP1 (ATCC^®^ TIB202™, Manassas, VA, USA), a monocytic non-adherent cell line from an acute monocytic leukemia was cultured in RPMI 1640 (Roswell Park Memorial Institute Medium 1640, Lonza, Basel, Switzerland), supplemented with 10% heat-inactivated fetal bovine serum (FBS Premium South America, PAN BIOTECH, Aidenbach, Germany); 5% L-glutamine (200 mM, Gibco, Thermo Fisher Scientific, Waltham, MA, USA); and 1% penicillin/streptomycin (10,000 U/mL/10,000 μg/mL, Gibco, Thermo Fisher Scientific). Cultures were maintained by the addition of a fresh medium or the replacement of the medium after centrifugation and kept at 5% CO_2_, 37 °C. Subcultures were proceeded to not exceed 1 × 10^6^ cells/mL.

### 4.4. PBMC Purification and Isolation

Buffy coat blood from a healthy anonymous donor (Croix Rouge, Service du sang, Suarlée, Belgium) was used for monocytes isolation. The formed elements of the blood were separated by a high-density gradient of Ficoll 1.077 g/mL (Lymphosep, VWR L0560-100, Nuaillé, France). Erythrocytes and granulocytes settled at the bottom of the tube while lymphocytes and monocytes remained at the sample–separation medium interface and the platelets were in the supernatant. The cell ring containing the Peripheral Blood Mononuclear Cells (PBMC) was collected. Then, the CD14+ monocytes were isolated from PBMC using CD14+ magnetic microbeads (Miltenyi Biotec, Leiden, The Netherlands). Separation columns (Miltenyi Biotec, Leiden, The Netherlands) were used to isolate the CD14+ monocytes with MACS MultiStand (Miltenyi Biotec, Leiden, The Netherlands) and MiniMACS Separator (Miltenyi Biotec, Leiden, The Netherlands) equipment. Monocytes from PBMC were then cultured in RPMI 1640 like THP1.

### 4.5. Macrophages Polarization

The THP-1 cells or monocytes from PBMC were differentiated in M0 macrophages during 24 h exposure with 25 ng/mL PMA (phorbol 12-myristate-13 acetate 1 mg, 8139 Sigma-Aldrich, St. Louis, MI, USA) in RPMI 1640, followed by a 72 h rest period in fresh media (RMPI 1640 without PMA) before treatment with polarizing cytokines. Adherent M0 macrophages were incubated for 24 h with 10 pg/mL LPS (lipopolysaccharides from Escherichia Coli O111:B4, Merck L2630-10 mg, Hoeilaart, Belgium) and 20 ng/mL INF-γ (recombinant Human IFN-gamma Protein, R&D Systems 285-IF-100, Minneapolis, MN, USA) to polarize in M1 macrophages. Alternatively, M0 macrophages can be polarized in the phenotype by the addition of 20 ng/mL IL-4 (human interleukin-4 protein 20 µg, ELL172, Interchim, Montluçon, France) and IL-13 (human interleukin-13 protein 20 µg, ELD140, Interchim, Montluçon, France) in RPMI 1640 and culture for 72 h.

### 4.6. RNA Extraction, cDNA Synthesis and qPCR

RNA extractions were performed on cell pellets with the InnuPrep RNA mini kit 2.0 (Annalytik Jena, Jena, Germany) following the recommendation of the provider. Quantification and purity verification of each extract was performed with the nanodrop (BioDrop μlite, Fisher scientific). Retro-transcription of RNA in cDNA was proceeded on 2 µg RNA using the Maxima First Strand cDNA Synthesis Kit for RT-qPCR with dsDNase (Thermo Scientific, K1671, Waltham, MA, USA). cDNAs were diluted 10-fold prior to being used for qPCR. Primer mixtures were prepared with RNAse-free water, 10 µM of the sense and nonsense primers (produced by IDT, Integrated DNA Technologies, Leuven, Belgium) (Table 2) and the SYBR green (Takyon Rox SYBR Core Kit Blue dTTP, Eurogentec, Selland, Belgium). Ten µL were dispensed into each well of a 96-well plate (Microplate for PCR, 96 wells, with sealing film, 732–1591 VWR) using a dispenser. Then, 2 µL of cDNA was added to each well. The plate was finally closed with a plastic film, centrifuged with a spinner, and introduced into the thermocycler (LightCycler 96 FW13083, Roche, Bâle, Switzerland). The program was launched according to the following cycle: preincubation 10′ at 95 °C, 2 steps amplification 15″ at 95 °C and 1′ at Tm (62 °C). This was repeated 40 times, followed by the melting curve 15″ at 95 °C, 1′ to Tm (62 °C), 95 °C continuous acquisition. Raw data are analyzed with LightCycler^®^ 96 SW 1.1 software. Data with a Cq error greater than 1 are discarded. Delta Ct, delta delta Ct and fold change (2^−ΔCt^) are calculated using the housekeeping gene 18S as a reference.

### 4.7. Immunofluorescence

Monocytes were polarized in macrophages on the coverslip of 24-well plates. When macrophages were differentiated in M1 and M2, cells were fixed for 15 min (10 min at 4 °C followed by 5 min at room temperature) with 4% paraformaldehyde (Sigma-Aldrich) in PBS, and then rinsed with PBS. For CD68 labeling, rinsing was followed by a permeabilization step in cold methanol for 10 min at −20 °C. Next, cells were incubated with a blocking solution, followed by antibody incubation (Table 3). After 3 washes, cells were incubated with the secondary antibody (1/500 dilution in blocking solution) for 1 h. The coverslips were then rinsed with PBS and distilled water before being mounted on slides with VectaShield-DAPI (VectaShield, Vector laboratories, Newark, CA, USA). Once dried, the slides could be observed under a confocal microscope (Nikon Ti2 A1RHD25, Tokyo, Japan).

### 4.8. Flow Cytometry

Briefly, after polarization in T75, M1 and M2 were collected and stained for surface markers during 10 min at 4 °C in DPBS/FBS 5% using CD86–APC (anti-human, monoclonal recombinant IgG1, REA968, Miltenyi Biotec, Leiden, The Netherlands); CD163–APC (anti-human, monoclonal recombinant IgG1, REA812, Miltenyi Biotec, Leiden, The Netherlands); and CD206–APC (anti-human, monoclonal recombinant IgG1, DCN228, Miltenyi Biotec, Leiden, The Netherlands) at a dilution of 1/50. For intracellular staining, cells were first fixed and permeabilized for 20 min at 4 °C with Cytofix/Cytoperm (BD Biosciences, Erembodegem, Belgium) and then washed with PermWash buffer (BD Biosciences, Erembodegem, Belgium). Next, cells were stained for 10 min at room temperature with CD68–PE antibody (anti-human, monoclonal recombinant IgG1, REA886, Miltenyi Biotec, Leiden, The Netherlands) diluted 50× in PermWash buffer. After washing, the cells were resuspended in DPBS/FBS 5% and analyzed by flow cytometry. The samples were assessed in BD LSRFortessa X-20 and the data was analyzed using FlowJo_v10.7.1.

### 4.9. ROS Evaluation

The THP1 monocytes were plated in a 25 cm^2^ flask and differentiated in macrophages. The detection of ROS in M1 and M2 macrophages was performed using the Muse^®^ Oxidative Stress Kit (Luminex, MCH100111, Austin, TX, USA). This kit is based on dihydroethidium (DHE), a permeable reagent used to detect ROS in cells. M1 and M2 macrophages were prepared in 1× assay buffer (Luminex, 4700–1330) at 3 × 10^6^ cells/mL. Then, 10 µL of cell suspension were incubated for 30 min at 37 °C with 190 µL of Muse^®^ Oxidative Stress Reagent (Luminex, 4700–1665, Austin, TX, USA) prediluted 800× with 1× assay buffer. Finally, the cell suspension was mixed thoroughly and run on the Guava^®^ Muse Cell Analyzer according to the manufacturer’s recommendations to detect ROS (−) and ROS (+) cells.

### 4.10. Statistical Analysis

Statistical analyses were performed using IBM SPSS Statistics software (version 21) (IBM, Ehningen, Germany). More than 2 independent samples were compared using an ANOVA test and a Tukey post-hoc test. For 2 independent samples, a T-test was used. A *p*-value <0.05 was considered as statistically significant (* = *p* ≤ 0.05; ** = *p* ≤ 0.01; *** = *p* ≤ 0.001). For all experiments, a minimum of 3 replicates were performed.

## 5. Conclusions

In our work, we indicate the high amount of M2 macrophages in HNSCC. We also report the possibility of targeting macrophages in HNSCC to set up new drugs in addition to the standard concomitant radio-chemotherapy, in order to improve cancer treatment. With our macrophage polarization protocol we can now study the efficacy of new molecules to reverse M2 to M1 macrophages. In this context, it is important to evaluate innovative macrophage reprogramming strategies that can also directly affect cancer progression. Regarding our results, targeting glutaminolysis and/or oxidative stress should be further examined. Moreover, working on freshly extracted monocytes from the blood of head and neck cancer patients will be possible to improve the clinical relevance of our model.

## Figures and Tables

**Figure 1 ijms-23-06385-f001:**
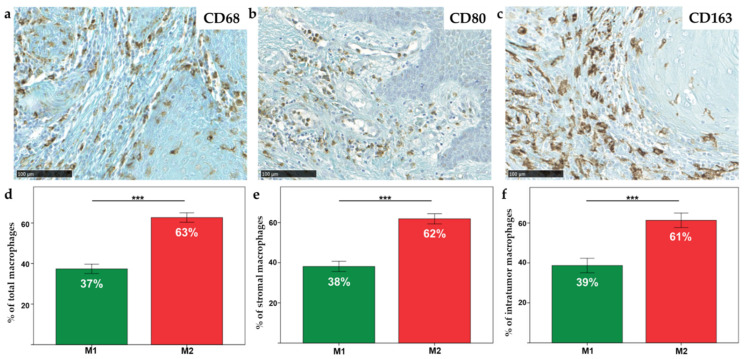
General view of (**a**) CD68, (**b**) CD80 and (**c**) CD163 immunohistochemical staining (scale = 100 µm); percentage of macrophages, M1 and M2 infiltration throughout (**d**) the tumor, (**e**) in ST and (**f**) IT compartments. *** = *p* ≤ 0.001.

**Figure 2 ijms-23-06385-f002:**
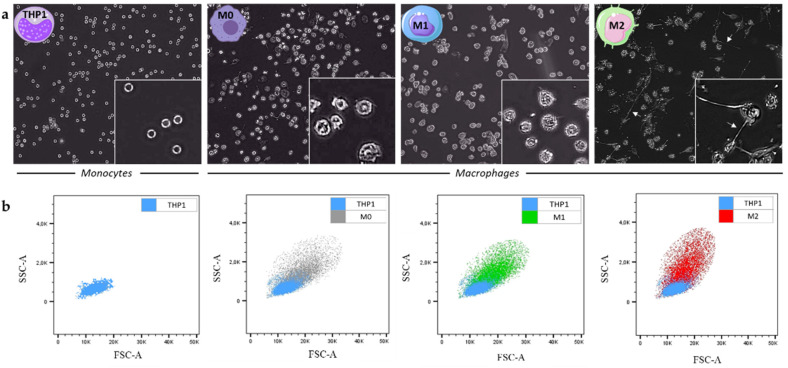
(**a**) Images of THP1 monocytes as well as the M0, M1 and M2 macrophage subtypes taken with the Euromex HD II microscope (magnification 40×) (zoomed in the bottom right of each photo, magnification 400×). The white arrows point to the cytoplasmic extensions characteristic of the M2 phenotype; (**b**) graphs of the granularity (SSC) of the polarized cells (M0 in grey, M1 in green and M2 in red) as a function of their size (FSC) were made using flow cytometry analyzes with the software FlowJo_v10.7.1, and compared with THP1 (in blue).

**Figure 3 ijms-23-06385-f003:**
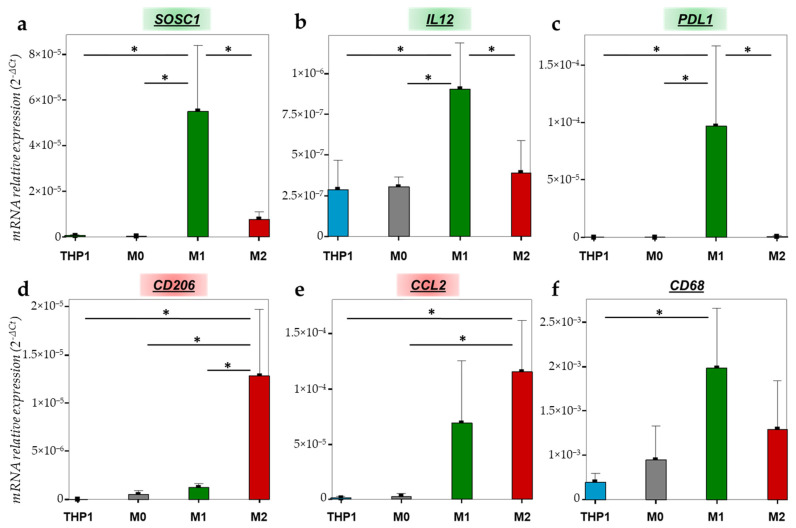
mRNA relative expression (2^−ΔCt^) of (**a**) *SOCS1*, (**b**) *IL12*, (**c**) *PD-L1*, (**d**) *CD206*, (**e**) *CCL2* and (**f**) *CD68* according to the cell type studied (THP1 monocytes (blue), M0 (grey), M1 (green) and M2 macrophages (red)), analyzed by RT-qPCR and normalized with *18S* expression. * = *p* ≤ 0.05.

**Figure 4 ijms-23-06385-f004:**
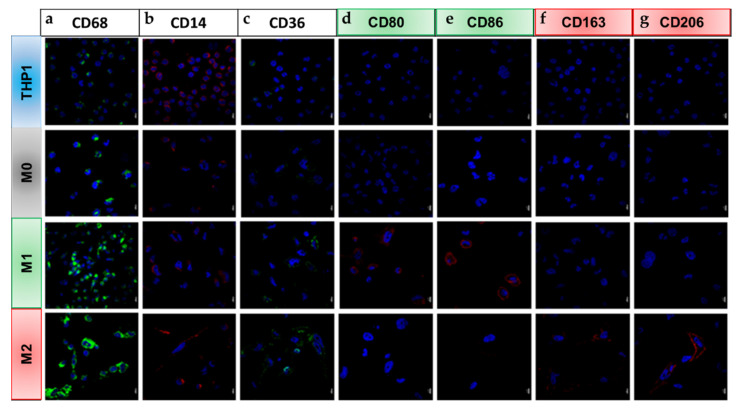
THP-1 monocytes differentiation in macrophages was observed by immunofluorescence. Cells were fixed and immuno-labeled for the monocytes and macrophages markers (**a**) CD68, (**b**) CD14 and (**c**) CD36; as well as M1 specific markers (**d**) CD80 and (**e**) CD86; and M2 markers (**f**) CD163 and (**g**) CD206 in the different cell types studied (THP1 monocytes (blue) or M0 (grey), M1 (green) and M2 macrophages (red). Nuclei were stained with DAPI (blue) (scale = 10 µm).

**Figure 5 ijms-23-06385-f005:**
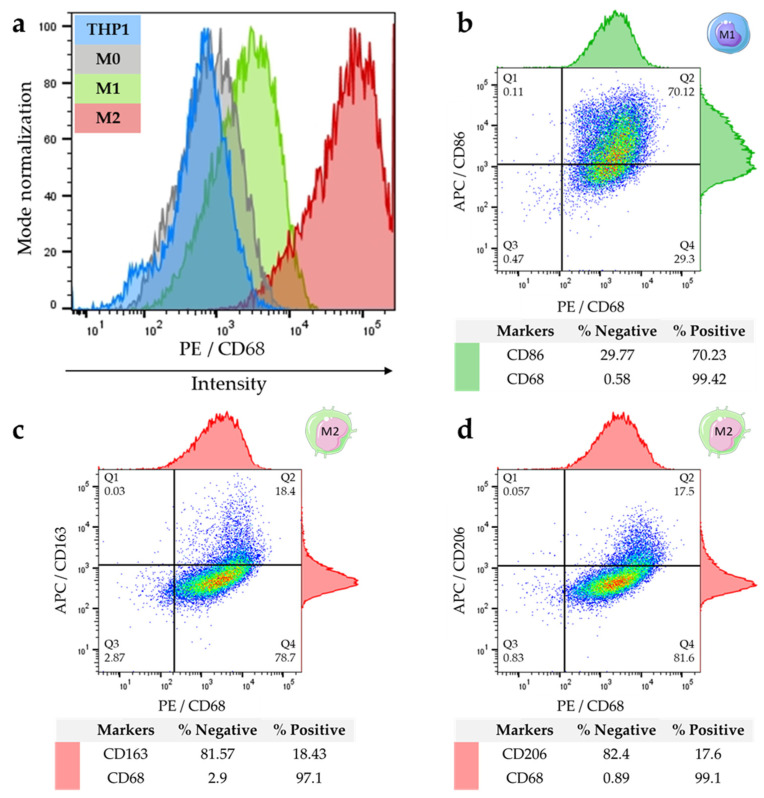
FACS analyses to compare CD68 expression in THP1 and macrophages, and to quantify the percentage of M1 and M2 macrophages expressing specific protein markers. (**a**) Cell surface CD68 intensity expression in THP-1 (blue), M0 (grey), M1 (green) and M2 (red); (**b**) dot plots of CD86/CD68 labeled in M1; (**c**) CD163/CD68 and (**d**) CD206/CD68 co-markers in M2. Analyses have been performed with the software FlowJo_v10.7.1.

**Figure 6 ijms-23-06385-f006:**
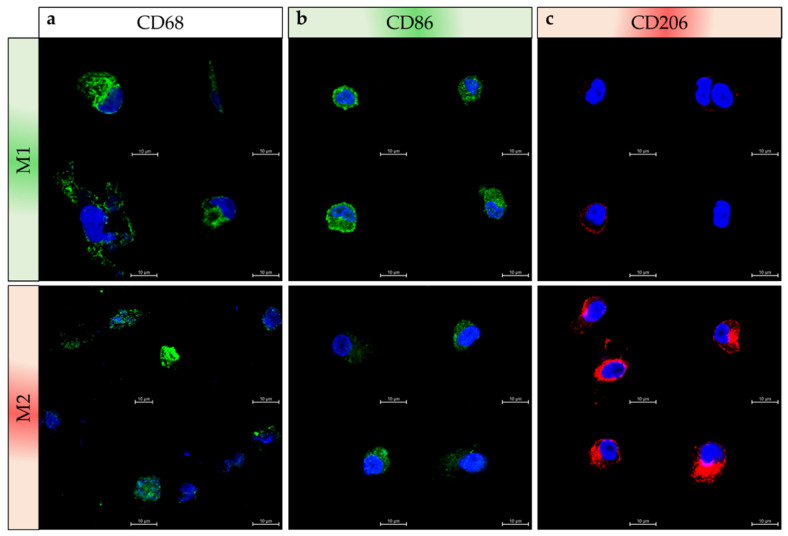
Differentiation of PBMC–derived monocytes in macrophages. Cells were fixed and immuno-labeled to detect (**a**) CD68, as well as (**b**) M1 specific marker CD86, and (**c**) M2 marker CD206 in the different cell types studied M1 (green) and M2 (red). Nuclei were stained with DAPI (blue) (scale = 10 µm). Four pictures are presented in each case of subtypes and immunostaining.

**Figure 7 ijms-23-06385-f007:**
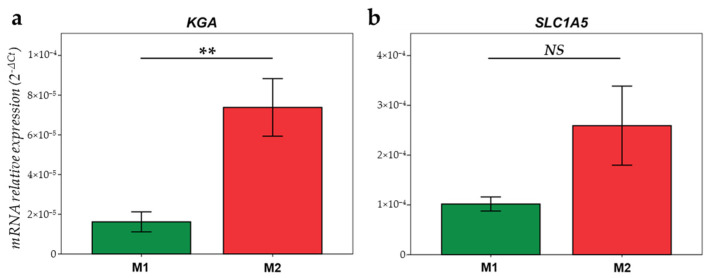
mRNA relative expression (2^−ΔCt^) of (**a**) *KGA* and (**b**) *SLC1A5* implicated in the glutamine metabolism has been studied by RT-qPCR in M1 (green) and M2 (red) macrophages and normalized with *18S* expression. *NS (non-significant)* = *p* > 0.05; ** = *p* ≤ 0.01.

**Figure 8 ijms-23-06385-f008:**
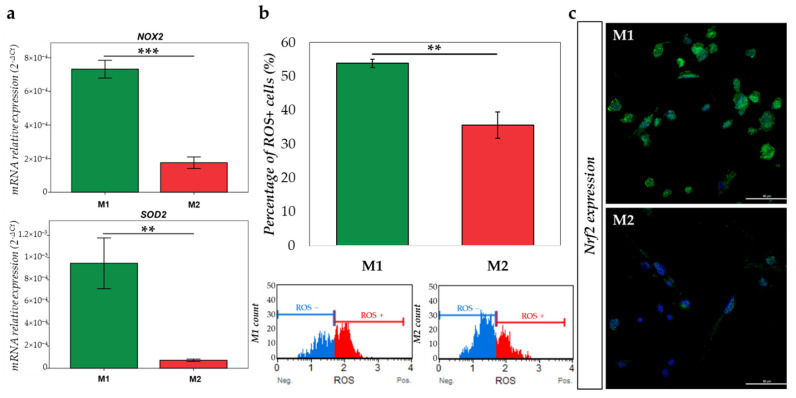
M1 and M2 oxidative stress comparison. (**a**) mRNA relative expression (2^−ΔCt^) of *NOX2* and *SOD2* involved in the regulation of oxidative stress has been studied by RT-qPCR in M1 (green) and M2 (red) macrophages and normalized with *18S* expression; (**b**) graph representing the percentage of ROS positive in M1 (green) and M2 (red) macrophages, plots show the histogram of gated cells with the 2 populations: ROS(−) (blue) and ROS(+) (red) cells; (**c**) immunofluorescence of Nrf2 (green) expression in M1 and M2 macrophages. Nuclei were stained with DAPI (blue) (scale = 50 µm). ** = *p* ≤ 0.01; *** = *p* ≤ 0.001.

**Figure 9 ijms-23-06385-f009:**
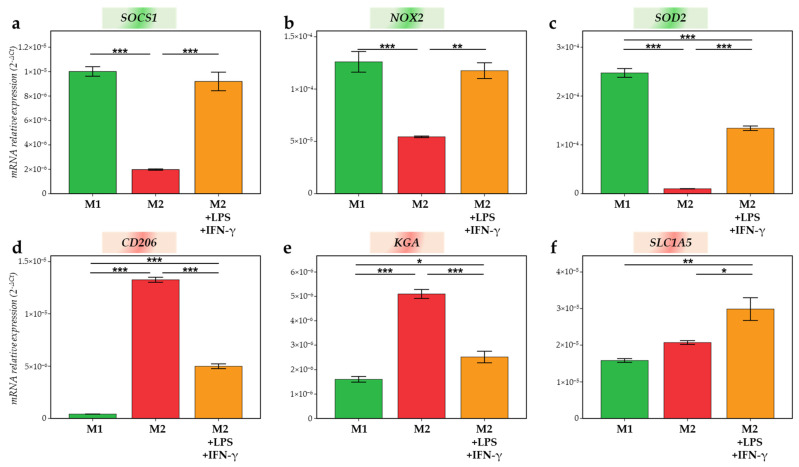
mRNA relative expression (2^−ΔCt^) of (**a**) *SOCS1*; (**b**) *NOX2*; (**c**) *SOD2*; (**d**) *CD206*; (**e**) *KGA*; and (**f**) *SLC1A5* studied by RT-qPCR in M1 (green), M2 (red) macrophages and M2 treated 24 h with 10 pg/mL LPS and 20 ng/mL IFN-γ (orange), normalized with *18S* expression. * = *p* ≤ 0.05; ** = *p* ≤ 0.01; *** = *p* ≤ 0.001.

**Table 1 ijms-23-06385-t001:** Patient population characteristics.

Variables	Number of Cases
	*n* = 60
**Age (years)**	
Median (range)	62 (42–89)
**Gender**	
Male	43
Female	17
**Anatomic site**	
Oral cavity	22
Oropharynx	19
Larynx	16
Hypopharynx	2
Nasopharynx	1
**Tumor stage**	
I-II	35
III-IV	18
Unknown	7
**Histological grade**	
Poorly differentiated	24
Well differentiated	30
Unknown	6
**Lymph nodes invasion**	
Yes	47
No	10
Unknown	3
**Risk factors**	
** *Tobacco* **	
Smoker	51
Non-smoker	9
** *Alcohol* **	
Drinker	37
Non-drinker	23
** *p16 status* **	
Positive	29
Negative	31
**Recurrence (RFS) (months)**	
Median (range)	14 (1–106)
Yes	26
No	32
Unknown	2
**Overall survival (OS) (months)**	
Median (range)	24 (1–294)
Alive	31
Dead	28
Unknown	1

**Table 2 ijms-23-06385-t002:** List of qPCR primers for human.

Genes	Forward Sequences	Reverse Sequences
*SOCS1*	TTTTCGCCCTTAGCGTGAA	CATCCAGGTGAAAGCGGC
*IL-12*	AAAATAGATGCGTGCAAGAGAGG	GGGGAAGACCTGTGACTTGAG
*PD-L1*	AAATGGAACCTGGCGAAAGC	GATGAGCCCCTCAGGCATTT
*CD206*	CTACAAGGGATCGGGTTTATGGA	TTGGCATTGCCTAGTAGCGTA
*CCL2*	CTCTCGCCTCCAGCATGAAA	TTTGCTTGTCCAGGTGGTCC
*CD68*	CTTCTCTCATTCCCCTATGGACA	GAAGGACACATTGTACTCCACC
*KGA*	GGTCTCCTCCTCTGGATAAGATGG	CCCGTTGTCAGAATCTCCTTGAGG
*SLC1A5*	TCATGTGGTACGCCCCTGT	GCGGGCAAAGAGTAAACCCA
*NOX2*	CCTAAGATAGCGGTTGATGG	GACTTGAGAATGGATGCGAA
*SOD2*	CACTGCAAGGAACAACAGGC	ACCAGGCTTGATGCACATCTT
*18S*	CATTTAGGTGACACTATAGAAGACGATCAGATACCGTCGTAGTTCC	GGATCCTAATACGACTCACTATAGGCCTTTAAGTTTCAGCTTTGCAACC

**Table 3 ijms-23-06385-t003:** Immunofluorescence conditions regarding primary antibodies.

Targets	Antibodies	Blocking Solutions	Primary Antibody Dilution and Incubation	Secondary Antibodies
CD68	Anti-human, rabbit monoclonal, anti-CD68, Cell Signaling	PBS/NGS 5%/Triton 0.3% (1 h)	1/800, overnight 4 °C	Goat anti-Rabbit IgG (H + L) Highly Cross-Absorbed Secondary Antibody, Alexa Fluor Plus 488
CD14	Anti-human, mouse monoclonal, anti-CD14, Miltenyi Biotec	PBS/BSA 2% (20 min)	1/100, overnight 4 °C	Goat anti-Mouse IgG (H + L) Highly Cross-Absorbed Secondary Antibody, Alexa Fluor Plus 555
CD36	Anti-human, rabbit monoclonal, anti-CD36, ThermoFisher Scientific	PBS/BSA 2% (20 min)	1/100, overnight 4 °C	Goat anti-Rabbit IgG (H + L) Highly Cross-Absorbed Secondary Antibody, Alexa Fluor Plus 488
CD80	Anti-human, mouse monoclonal, anti-CD80, R&D Systems	PBS/casein 0.5% (1 h)	1/50, overnight 4 °C	Goat anti-Mouse IgG (H + L) Highly Cross-Absorbed Secondary Antibody, Alexa Fluor Plus 555
CD86	Anti-human, rabbit monoclonal, anti-CD86, Cell Signaling	PBS/NGS 5%/Triton 0.3% (1 h)	1/100, overnight 4 °C	Goat anti-Rabbit IgG (H + L) Highly Cross-Absorbed Secondary Antibody, Alexa Fluor Plus 488
CD163	Anti-human, mouse monoclonal, anti-CD163, Sanbio	PBS/BSA 2% (20 min)	1/50, overnight 4 °C	Goat anti-Mouse IgG (H + L) Highly Cross-Absorbed Secondary Antibody, Alexa Fluor Plus 555
CD206	Anti-human, mouse monoclonal, anti-CD206, Miltenyi Biotec	PBS/BSA 2% (20 min)	1/100, overnight 4 °C	Goat anti-Mouse IgG (H + L) Highly Cross-Absorbed Secondary Antibody, Alexa Fluor Plus 555
Nrf2	Anti-human, rabbit monoclonal, anti-Nrf2, Cell Signaling	PBS/NGS 5%/Triton 0.3% (1 h)	1/200, overnight 4 °C	Goat anti-Rabbit IgG (H + L) Highly Cross-Absorbed Secondary Antibody, Alexa Fluor Plus 488

## Data Availability

Data is contained within the article or supplementary material.

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
