# Peer review of "Dealing with Macrophage Plasticity to Address Therapeutic Challenges in Head and Neck Cancers"

_ijms, 2022, doi:10.3390/ijms23126385_

Round 1

Reviewer 1 Report

This study evaluates the tumor-associated macrophage plasticity in Head and Neck Cancers to find new therapeutic approaches. The targeting of M2 macrophages to repolarize into the M1 phenotype is the hypothesis from which the study starts. The concept of this study is very interesting in that after the authors developed a protocol for reversible polarization of macrophages, they tried to validate it by analyzing the expression of some genes and proteins involved in this process.

 ROS production is required for macrophages M1 activation and intervention. Furthermore, the oxidative stress pathways are important for macrophage polarization, and for this, the production of ROS in M1 macrophages was analyzed compared to M2. The results led to the observation that by modulating ROS levels in the tumor, macrophages can be reprogrammed while targeting cancer cells.

The conclusions are supported by the data presented, this paper being of interest considering the original approach on the very hot topic addressed.

I would have some suggestions for improving the graphical representation of some figures:

  • In Figure 3. the values on the y-axis are difficult to read; In the legend, delete ** = 184 p≤0.01; *** = p≤0.001 because they do not appear in graphics
  • In Figure 7. delete * = p≤0.05; *** = p≤0.001; it must be explained what NS means
  • In Figure 8:

a-   the values on the y-axis are difficult to read;

b-   the values on the y axis must be changed (10,20, 30… .50)

  • In Figure 9. the values on the y-axis are difficult to read.

This article is well designed, uses recent literature data, and can be interesting for readers.

There are some typographical and grammatical errors in the manuscript that I have highlighted.

Overall, I consider the article could be a useful contribution to the journal. I recommend the manuscript for publishing after minor changes and updates have been taken into consideration by the authors.

Reviewer 2 Report

This manuscript entitled Dealing with Macrophage Plasticity to Address Therapeutic Challenges in Head and Neck Cancers evaluated macrophage infiltration in 60 HNC patients and found this high infiltration of CD68+ cells mainly related to CD163+ M2 macrophages. The study optimized a polarization protocol from THP1 monocyte cell line and demonstrated the plasticity of this model by using M1 activators to repolarize M2 into M1 and concludes that the study could provide a complete reversible polarization protocol allowing to further evaluate various reprogramming effectors targeting glutaminolysis and/or oxidative stress in macrophages.

This study has high originality and novelty with high quality of presentation. There are two comments and suggestions:

1.Regarding tumor progression is associated with an increase in the M2/M1 ratio and is related to a poor therapeutic outcome (references 24, 25), suggest to add the data of M2/M1 ratio in patients with tumor (26 patients) or without tumor (32 patients) recurrence and compare the impact on survival;

2.Suggest to indicate the source of PBMC in Materials and Methods 4.4. PBMC Purification and Isolation, from the healthy donor or from HNC patients?

3. Line 133 Figure 2d should be corrected as Figure  1d ; Line 134, Figure 2e be corrected as 1e; Line 135, Figure 2f be corrected as 1f.

Reviewer 3 Report

This is an interesting study about macrophage plasticity and therapeutic challenges in head and neck cancer.

The paper is well written. However, some issues remain.

Since nasopharyngeal carcinoma is usually related to Epstein-Bar Virus, I think that such case should be excluded from the study.

The authors must explain what “tumor invasion” means (all tumors are invasive).

Correlation analyses between M1/M2 infiltrate and clinical data may be interesting.

Reviewer 4 Report

In this article, the authors check the polarization of macrophages in head and neck tumor microenvironment from HNC patients. The major population of these macrophages is CD163+CD68+ M2-like macrophages. The authors further investigated the metabolic features of these cells, glutaminolysis (M2), and oxidative stress in (M1).  However, these are typical features of macrophages in M1 or M2 polarization.

Figure 1d-f, what is %age? Which tumor section is it from?

Figure 3, why not check the CD163 expression?

Figure 6, where are the PBMCs from?

The stimulation of M1 or M2 macrophages should be performed using tumor lysate, conventional stimulation should be only used as control.

Round 2

Reviewer 3 Report

Thanks for your response.

Author Response

Dear Reviewer,

Thank you.

Reviewer 4 Report

For Figure 1d-f, do not use %age for y-axis title,  please change  % or Frequency of total macrophages; % of stromal macrophages; % of intratumor macrophages.

Author Response

Dear Reviewer,

The y-axis title in the Figure 1d-f were modified as requested.

Thank you for your comments.